# Skymask Matching Aided Positioning Using Sky-Pointing Fisheye Camera and 3D City Models in Urban Canyons

**DOI:** 10.3390/s20174728

**Published:** 2020-08-21

**Authors:** Max Jwo Lem Lee, Shang Lee, Hoi-Fung Ng, Li-Ta Hsu

**Affiliations:** Interdisciplinary Division of Aeronautical and Aviation Engineering, The Hong Kong Polytechnic University, Hong Kong SAR 999077, China; 16044209d@connect.polyu.hk (M.J.L.L.); 16044131d@connect.polyu.hk (S.L.); ivannhf.ng@connect.polyu.hk (H.-F.N.)

**Keywords:** GPS, GNSS, localization, navigation, autonomous driving, urban canyon, land application, cameras, image segmentation

## Abstract

3D-mapping-aided (3DMA) global navigation satellite system (GNSS) positioning that improves positioning performance in dense urban areas has been under development in recent years, but it still faces many challenges. This paper details a new algorithm that explores the potential of using building boundaries for positioning and heading estimation. Rather than applying complex simulations to analyze and correct signal reflections by buildings, the approach utilizes a convolutional neural network to differentiate between the sky and building in a sky-pointing fisheye image. A new skymask matching algorithm is then proposed to match the segmented fisheye images with skymasks generated from a 3D building model. Each matched skymask holds a latitude, longitude coordinate and heading angle to determine the precise location of the fisheye image. The results are then compared with the smartphone GNSS and advanced 3DMA GNSS positioning methods. The proposed method provides degree-level heading accuracy, and improved positioning accuracy similar to other advanced 3DMA GNSS positioning methods in a rich urban environment.

## 1. Introduction

Global navigation satellite systems (GNSS) provide geographical longitude and latitude positioning with meter-level accuracy in open areas [1]. This accuracy, however, suffers in dense urban areas because buildings block, reflect, and diffract the signals. These cause errors in satellite positioning and reduces accuracy and, in severe cases, the position error could exceed 50 m [2,3]. An improvement in the real time-positioning accuracy of low-cost GNSS systems in dense urban areas to within 5 m would benefit many different potential applications [4], such as cloud-sourced mobile mapping and object tracking. As such, there is a need for a low-cost positioning device that has great solution availability and accuracy.

To improve the positioning performance in urban environments, researchers have designed different methods to identify then correct or exclude the unhealthy measurements by receiver based GNSS or with extra equipment. The increased availability of open-sourced 3D building models allows utilization of 3D-mapping-aided (3DMA) GNSS positioning to improve urban positioning [4,5]. This includes shadow matching, ray tracing, and likelihood-based ranging methods. Shadow matching utilizes 3D building models with building geometry to match satellite visibility [1,6], allowing the exclusion of NLOS measurements [7]. Ray tracing [8,9,10] predicts the signal transmission path based on building geometry to calculate the reflection delay distance and provide correction on the pseudorange of the NLOS signals [8,9,10]. The likelihood-based ranging method is another pseudorange correction method, in which the GNSS position is corrected via a statistical model [11]. In addition, the satellite visibility can be used to estimate the sky-visibility of the environment where the receiver is located [12].

Another approach towards enhancing positioning accuracy in the urban environment is using extra equipment to collect additional data for positioning. This approach is most suitable for vehicular applications due to weight, space, and power usage concerns. Two popular approaches include the use of 3D light detection and ranging (LiDAR) and the usage of fisheye cameras. LiDAR is used to retrieve surrounding building or obstacle information, which in turn can be used to perform NLOS classification [13]. With the ability to obtain the building distance, the pseudorange can be corrected by the NLOS propagation model [14]. Sky-pointing fisheye cameras are capable of detecting obstacles and buildings in the local environment. When used in conjunction with image processing algorithms [2,4,14,15,16,17], they allow the exclusion of NLOS satellites from position calculations, improving positioning accuracy [15]. Furthermore, research indicates that if both LiDAR and fisheye cameras are used in conjunction, positioning accuracy in urban areas can be further enhanced [3].

Fisheye cameras are already used in autonomous driving, where a front-facing camera is commonly used for lane detection and road sign identification. To facilitate its use in improving GNSS positioning performance, the fisheye camera can be pointed towards the zenith, which reduces the probability of capturing uncertainties while allowing the use of the positions of immovable objects such as high-rise buildings. As such, this paper makes use of images captured from a zenith sky-pointing fisheye camera, which are matched to computer-generated boundary skyplot (skymask) to obtain a position and heading. This is similar to the approach employed in the research paper [18]. However, this study differs in several ways. Firstly, in this research paper, the roll and pitch are fixed with a fisheye camera pointing upward, hence the 3D model can be used to generate the skymask ahead of time, while the aforementioned paper utilizes real-time generation of computer-generated images from a 3D model. This study’s method uses less computing power, while the [18] study’s method allow generation of images better suited to the camera’s condition (pitch, roll, etc.). This study also utilizes a convolutional neural network (CNN) to segment images into sky and building classes, while the study by Suzuki and Kubo [18] uses the Otsu method to differentiate between sky and building. Semantic segmentation was chosen for this study due to its potential to segment additional classes in the future. Lastly, the proposed matching algorithm can not only provide the position of the user but also the heading angle. These are the main novelties of this research paper.

The proposed skymask matching method integrates GNSS, supervised deep learning, and a matching algorithm. This approach takes advantage of sky-pointing fisheye images and matching it with a candidate skymask. The results are then compared with other enhanced methods of positioning. The rest of the paper is organized as follows. The proposed algorithm will be explained in detail in Section 2. Section 3 describes the experimentation results. Section 4 contains the concluding remarks.

## 2. Proposed Skymask Matching Method

An overview of the skymask matching method is shown in Figure 1. This study proposes the calculation of the position solution and heading solution with sky-pointing fisheye image and 3D building model generated skymask. The method is divided into two main categories, an offline process and an online process. The offline process is done ahead of real-time processing. It consists of training the CNN and generating 3D building model skymasks, which are generated outside of buildings in square grid intersections, with a length and width separation of 2 m. In the online process, the user captures a sky-pointing fisheye image, with initial position and heading estimated by a smartphone GNSS receiver. Then, 3D building model candidate positions are distributed around the initial positioning in a 50 m radius to generate candidate skymasks. Meanwhile, the image is simultaneously segmented using the convolutional neural network to distinguish between building and sky, which is then converted to a binary image. Then, the binary image is converted from pixel format to azimuth and elevation format for comparison. The image-converted skymask is compared to candidate skymasks using Global Matching Simple Difference (GMSD), Global Matching Standard Deviation Difference (GMSDD), Feature Matching Simple Difference (FMSD), Feature Matching Breaking Point Difference (FMBPD), and Heading Difference (HD) GMSD techniques. The techniques were weighted and combined into a score to calculate the likelihood of the candidates. The chosen position and heading solution are determined by the highest combined score. 

### 2.1. Skymask Generation

In this paper, the sky-pointing fisheye image is compared to precompute skymasks that are identical to those utilized in shadow matching and skymask 3DMA GNSS. Skymasks store the highest elevation angle of the surrounding building boundaries (where buildings meet the sky) at each azimuth angle. When used in skymask matching, these skymasks are generated in the offline stage and stored in a database, an example is shown above in Figure 1. As the offline stage is done ahead of time, skymask matching requires less computing power and reduces processing times by a factor of 10 when compared to the real-time use of the 3D building model [19]. The precomputed skymasks are then used in the online phase for matching. The details of the generation processes are listed in [15]. Each skymask has a resolution of 360 azimuth angles with one degree resolution, and 90 elevation angles with 0.1 degree resolution. They also contain the WGS84 latitude and longitude of the location in decimal degrees format.

### 2.2. Semantic Segmentation of the Image Taken by A Sky-Pointing Camera

Preprocessing is needed before matching the fisheye camera image with the candidate skymask in the online stage. There are two main steps for the image processing used: (1) image segmentation by a neural network; (2) image coordinate conversion from pixel to elevation and azimuth angles. These will be discussed in Section 2.2.1 and Section 2.2.2, respectively.

#### 2.2.1. Convolutional Neural Network (CNN) Training

Training datasets were collected at deep urban canyons in Hong Kong. The locations were chosen due to their challenging urban topography. The high density of skyscrapers and other tall buildings create steep urban canyons, reducing the accuracy of GNSS positioning. The high concentration of skyscrapers also leads to an abundance of features in a sky-pointing image, providing an ideal testing ground for skymask matching.

Over 1200 daytime images were taken using a Canon DSLR camera. Of these, more than 570 images were manually labelled and used to train the CNN semantic segmentation network. The dataset is split into three parts: 60% are used as a training set, 20% as the cross-validation set, and the remaining 20% as the test set. The training dataset was labelled manually with the Image Labeler application in MATLAB, a software created and maintained by Mathworks of Natick, Massachusetts, USA. This Image Labeler application is part of the Computer Vision Toolbox [20]. MATLAB’s Deep Learning toolbox was also utilized to retrain Resnet50, a 50-layer CNN [21]. The trained CNN is then used for semantic segmentation. In this paper, semantic segmentation is used to differentiate between the sky and buildings only. In the future, however, more classes could be added. This will be discussed in further detail in Section 4.

To further improve the segmentation accuracy, active contours are also applied. Active contouring utilizes the Chan-Vese segmentation algorithm [22]. The Chan-Vese method is based on the approach of curve evolution to separate the foreground from background based on the means of two regions.

#### 2.2.2. Image Coordination Transformation: Pixel to GNSS Skyplot 

To match the segmented fisheye image with the precomputed skymask, the segmented image is first converted into a format that is identical to that of the precomputed skymask, known as the GNSS skyplot format (elevation and azimuth angles). The resolution of elevation and azimuth angle information are identical to the precomputed skymask in Section 2.1. For a given position in pixel, the calculation to convert into azimuth and elevation angle is the same approach described in [3]. Assuming the optical center of the camera is zenith pointing, each pixel inside the sky view image will be converted to a corresponding azimuth and elevation angle.
(1)$$\theta_{z}=\frac{\pi }{2}-el_{z}^{\mathrm{image}};\;\;d_{z,\mathrm{pix}}=2\cdot f_{c}\tan\left(\frac{\theta_{z}}{2}\right),z\in \left\{index\;of\;pixels\right\}$$

To determine the elevation angle for a pixel ($x_{z,\mathrm{pix}}^{\mathrm{image}},y_{z,\mathrm{pix}}^{\mathrm{image}})\;$ the focal length ($f_{c}$) of the fish-eye camera is needed. Where $\;d_{z,\mathrm{pix}}$, is the pixel distance from the center of the sky-pointing image and correlated with the elevation angle $el_{z}^{\mathrm{image}}$ of the pixel. Given the center of the sky-pointing image at pixel position ($x_{c,\mathrm{pix}}^{\mathrm{image}},\;y_{c,\mathrm{pix}}^{\mathrm{image}})$, the azimuth and elevation angle can be expressed as follows:(2)$$x_{z,\mathrm{pix}}^{\mathrm{image}}=x_{c,\mathrm{pix}}^{\mathrm{image}}+d_{z,\mathrm{pix}}\cdot \cos\left(az_{z}^{\mathrm{image}}\right);\;\;y_{z,\mathrm{pix}}^{\mathrm{image}}=y_{c,\mathrm{pix}}^{\mathrm{image}}-d_{z,\mathrm{pix}}\cdot \sin\left(az_{z}^{\mathrm{image}}\right)$$

The expression is a quadratic equation that solves the azimuth and elevation angles simultaneously. For practicality, the relationships between the angles and corresponding pixel coordinates are precomputed offline and stored in a database. This means that during the online matching process, the angles’ information can be retrieved by mapping given pixel coordinates in the precompute lookup table, reducing computational load and time.
(3)$$M_{z,\mathrm{pix}}^{\mathrm{image}}=\left[az_{0}^{\mathrm{image}},el_{0}^{\mathrm{image}}\ldots az_{z}^{\mathrm{image}},el_{z}^{\mathrm{image}}\right]$$

### 2.3. Skymask Matching Positioning and Heading Resolution

In the online stage, the sky-pointing fisheye image-converted skymask is rotated 359 times (j) with an increment of one degree clockwise to generate 360 skymasks, each with a different heading angle. The image-converted skymask is then compared to the skymask generated from each candidate position (k). The matching algorithm will provide two pieces of information: (1) the location of the fisheye camera, and (2) the heading angle of the camera. The target function is to find the candidate skymask with the smallest differences with respect to elevation angle, standard deviation, feature, breaking points, and heading angle.
(4)$$x_{k}=\left[p_{k}^{lat},p_{k}^{lon}\right],k\in \left\{distributed\;candidates\right\};\;el=\left[el_{0},\ldots ,el_{az},\ldots ,el_{359}\right],\;az=\left[0{}^{\circ},359{}^{\circ}\right];\;\;el_{k}^{3D\;\mathrm{model}}=\mathrm{skymask}\left(x_{k}\right)\;;\;el_{j}^{\mathrm{image}}=\mathrm{image}\left(\psi_{j}\right)\;,\;j=\left[0{}^{\circ},359{}^{\circ}\right]$$

There are five methods to compare two skymasks, which are (1) Global Matching Simple Difference (GMSD), (2) Global Matching Standard Deviation Difference (GMSDD), (3) Feature Matching Simple Difference (FMSD), (4) Feature Matching Breaking Point Difference (FMBPD), and (5) Heading Difference (HD). In total, five sets of differences can be obtained to use for the estimation of the likelihood of the distributed candidates.

#### 2.3.1. Global Matching Simple Difference (GMSD)

At each azimuth angle, the difference of elevation angles between the image-converted skymask, and the skymask at different candidates based on 3D model is calculated and averaged. The averaged elevation angle difference will be known as the $gmsd_{j.k}^{\mathrm{image},\;3D\;\mathrm{model}}$ value.
(5)$$gmsd_{j.k}^{\mathrm{image},3D\;\mathrm{model}}=\frac{\left|el_{j}^{\mathrm{image}}-el_{k}^{3D\;\mathrm{model}}\right|}{360}$$

The average elevation angle difference is obtained by summing the absolute elevation angle difference at each azimuth angle. This difference is summed up and divided by 360 (the number of azimuths) to acquire the average elevation angle difference. A large average difference means the candidate skymask has a low probability to be the image-converted skymask, whereas a smaller value represents a similar overall average elevation difference, and thus a higher probability of the candidate skymask being similar to the image-converted skymask.

#### 2.3.2. Global Matching Standard Deviation Difference (GMSDD)

The second technique is GMSDD. At each azimuth, the elevation angle is compared to the average $\bar{el_{j}^{\mathrm{image}}}$ and $\bar{el_{k}^{3D\;\mathrm{model}}}$ to calculate its deviation from the mean. The standard deviation algorithm is used to measure the variation of elevation in specified skymask. A small standard deviation indicates little to no variation in elevation. A large standard deviation shows a large variation in elevation. The standard deviation of the candidate skymask is then compared to the standard deviation of the image-converted skymask. A smaller difference means higher similarity.
(6)$$gmsdd_{j.k}^{\mathrm{image},3D\;\mathrm{model}}=\sqrt{\frac{{\left(el_{j}^{\mathrm{image}}-\bar{el_{j}^{\mathrm{image}}}\right)}^{2}}{360}}-\sqrt{\frac{(el_{k}^{3D\;\mathrm{model}}-{\bar{el_{k}^{3D\;\mathrm{model}})}}^{2}}{360}}$$

#### 2.3.3. Feature Matching Simple Difference (FMSD)

The third method measures the features only. Features are defined as changes in elevation angle between two adjacent azimuth points, an example of which can be seen in Figure 2. Algorithm 1 shows the feature identifying process in a skymask.

**Algorithm 1** Determining the Feature Points in a Skymask**Input**:   Skymask, el**Output**:  Skymask Feature Points, el′ 1:   **for** each azimuth, az in el2:    calculate az and az+1 elevation angle difference, β3:    update el′(az)=β4:   end for each azimuth, az

The average feature difference is obtained in a similar way to GMSD, through summing the absolute feature difference at each azimuth angle. This difference is summed up and divided by 360 to acquire the average feature difference. A smaller FMSD score represents a similar overall feature and, thus, a higher similarity between the candidate skymask and image-converted skymask.
(7)$$fmsd_{j.k}^{\mathrm{image},3D\;\mathrm{model}}=\frac{\left|el_{j}^{\mathrm{image}\prime }-el_{k}^{3D\;\mathrm{model}\prime }\right|}{360}$$

#### 2.3.4. Feature Matching Breaking Point Difference (FMBPD)

The breaking points are defined as the features greater than 10 elevation difference in a skymask, an example can be seen in Figure 2. They are calculated for both image-converted skymask $\left(el_{j}^{\mathrm{image}}\right)$ and candidate skymask $\left(el_{k}^{3D\;\mathrm{model}}\right)$. By using the extremly distinctive features on a skymask only, a fourth FMBPD score can be obtained by both skymask with breaking point estimation shown in Algorithm 2.

**Algorithm 2** Determining the Breaking Points in a Skymask.**Input:**   Skymask Feature Points, el’**Output:**  Skymask Breaking Points, elbreak′ 1   elbreak′=[360∗1] zeros vector2   **for** each azimuth, az in el′3    **if**
el′(az) > 10 elevation difference, βbreak4    update elbreak′(az)=βbreak5    **end if**6   **end for** each azimuth, az

The FMBPD can be calculated by taking the difference of the breaking points between the image-converted skymask and candidate skymask demonstrated in Algorithm 3. In our experience of matching the image-converted skymask with its respective 3D model ground truth skymask, the breaking points are within a two degree azimuth offset. Therefore, if the breaking points of an image-converted skymask is within two azimuth degrees of the candidate skymask, a score can be obtained.

**Algorithm 3** Feature Matching Breaking Point Difference between Two Skymasks.**Input:**   Image-converted skymask Breaking Points, eljimage, break′     Candidate Skymask Breaking Points, elk3D model,break′**Output:**  Feture Matching Breaking Point Difference, fmbpdj.kimage, 3D model 1   **for** each azimuth, az2    **if**
eljimage, break′(az)>0,  βimage,  break3     **if**
elk3D model,break′(az−2 or az−1 or…or az+2)>0,  β3D model,  break.4      update fmbpdj.kimage, 3D model=|βimage,  break−β3D model,  break|5     **end if**6    **end if**7   **end for** each azimuth, az

Candidates with no FMBPD will have no score, a smaller FMBPD value represents a similar overall feature and, thus, a higher similarity between the candidate skymask and image-converted skymask.

#### 2.3.5. Heading Difference (HD)

The fifth formula calculates the heading difference between the rotated image-converted skymask and the smartphone heading, which is the heading recorded by the used in this study. A smaller heading difference from the smartphone heading of the image-converted skymask when matched with a candidate skymask means they are more similar.
(8)$$\varphi =\psi_{j,k}^{\mathrm{image},3D\;\mathrm{model}}-\psi_{j}^{\mathrm{smartphone}};\;hd_{j,k}^{\mathrm{image},\;3D\;\mathrm{model}}=\{\begin{array}{cc} \varphi +360^{o} & \varphi <-180^{o}\\ \varphi -360^{o} & \varphi >180^{o}\\ \varphi & -180^{o}\leq \varphi \leq 180^{o} \end{array}$$

### 2.4. Candidate Scoring

A higher score is given to the candidate position with a higher similarity between the image-converted skymask and the candidate skymask. Gaussian distributions are assumed and used to model the the similarity of the candidate skymasks. In theory, the sky-pointing fisheye image taken at the corresponding computer-generated GT skymask should have the smallest difference. Ten sky-pointing fisheye images are taken at the corresponding known GTs to calibrate the Gaussian probability distribution function (PDF). The five differences are used to calculate the corresponding probability value in their respective distributions. The combined likelihood becomes the weighting score of each candidate.
(9)$$s_{gmsd,j,k}^{\mathrm{image},3D\;\mathrm{model}}=\frac{1}{8*\sqrt{2\pi }}e^{-\frac{1}{2}{\left(\frac{gmsd_{j.k}^{\mathrm{image},3D\;\mathrm{model}}}{8}\right)}^{2}};\;s_{gmsdd,j,k}^{\mathrm{image},3D\;\mathrm{model}}=\frac{1}{15*\sqrt{2\pi }}e^{-\frac{1}{2}{\left(\frac{gmsdd_{j.k}^{\mathrm{image},3D\;\mathrm{model}}}{15}\right)}^{2}};\;s_{fmsd,j,k}^{\mathrm{image},3D\;\mathrm{model}}=\frac{1}{15*\sqrt{2\pi }}e^{-\frac{1}{2}{\left(\frac{fmsd_{j.k}^{\mathrm{image},3D\;\mathrm{model}}}{15}\right)}^{2}};\;s_{fmbpd,j,k}^{\mathrm{image},3D\;\mathrm{model}}=\frac{1}{20*\sqrt{2\pi }}e^{-\frac{1}{2}{\left(\frac{fmbpd_{j.k}^{\mathrm{image},3D\;\mathrm{model}}}{20}\right)}^{2}};\;s_{hd,j,k}^{\mathrm{image},3D\;\mathrm{model}}=\frac{1}{40*\sqrt{2\pi }}e^{-\frac{1}{2}{\left(\frac{hd_{j.k}^{\mathrm{image},3D\;\mathrm{model}}}{40}\right)}^{2}};\;s_{combined,j,k}^{\mathrm{image},3D\;\mathrm{model}}=s_{gmsd,j,k}^{\mathrm{image},3D\;\mathrm{model}}+s_{gmsdd,j,k}^{\mathrm{image},3D\;\mathrm{model}}+s_{fmsd,j,k}^{\mathrm{image},3D\;\mathrm{model}}+s_{fmbpd,j,k}^{\mathrm{image},3D\;\mathrm{model}}+s_{hd,j,k}^{\mathrm{image},3D\;\mathrm{model}}$$

Finally, the candidate skymask with the largest combined score will be selected as the chosen candidate skymask. The combined score will then be normalized and rescaled between 0 and 100%, which represents the total score of the candidate. Estimated heading angle offset corresponds to the chosen candidate skymask.

## 3. Experiments Results and Analysis

### 3.1. Experiment Setup

In this study, the experiment locations were selected within the Tsim Sha Tsui area of Hong Kong, as shown in Figure 3. These locations were selected using the following factors: proximity to obstacles, the height and features of nearby buildings, and the ability to determine the location relative to landmarks both on the ground and by satellite imagery. Images were then taken at each of the selected ground truths using a digital single-lens reflex (DSLR) camera. The DSLR (Canon 5D Mk III DSLR) with the fisheye-lens (8–15 mm f/4L EF fisheye USM lens). Both the lens and DSLR were designed and manufactured by Canon Inc., a company based in Ōta, Tokyo, Japan. The camera was used to capture the image, while a Samsung Galaxy Note 9 smartphone was used to record the low-cost GNSS solutions and heading. The GNSS receiver within the Note 9 was a Broadcom BCM47755, which was designed by Broadcom Corporation, a company based in Irvine, California, USA. The images taken were then manually categorized into four distinctive environments, with distinctions specified in Table 1 below. Categorization was based on the frequency of different obstacles, buildings, and its features. Four images were chosen, one from each category, to demonstrate the proposed algorithm.

The experimental results are then post-processed and compared to the ground truth and different positioning algorithms, including:(1)Skymask Matching (SM): The skymask matching algorithm proposed in this paper;(2)Smartphone: Low-cost GNSS solution by Galaxy note 9 Broadcom BCM47755;(3)3DMA: Integrated solution by 3DMA GNSS algorithm on shadow matching, skymask 3DMA and likelihood-based ranging GNSS [23];(4)WLS: Weighted-least-square [24];(5)Conv: Conventional commercial GNSS solution by Allystar TAU1302+AGR6303 active antenna;(6)Ground Truth: Data was collected at the landmark location on Google Earth, a software maintained and developed by Google LLC, which is based in Mountain View, California, USA. The accuracy is within 1–2 m based on our experience.

The images used for testing were not used in training the CNN. In the images shown in Table 2, the heading of the camera (and thus the fisheye image) faces north at 0 degree in azimuth angle. During the image collection process, these images were aligned to true north. North was determined by estimation using Google Earth and observations of nearby objects. Slight manual adjustments ± 2° to the heading were made after the images were taken to ensure heading of the image faces true north for verification purposes. These adjustments were also determined based on Google Earth. It is important to note that the heading estimated by the smartphone was also adjusted to the same magnitude for consistency.

To evaluate the accuracy of the segmentation processing on the images, they were compared to their corresponding hand-labelled counterparts. As seen in Table 2, four images were tested to see if the proposed method provides a consistent segmentation to differentiate between buildings and sky. 

### 3.2. Evaluation of the Skymask Generated Based on Image and 3D Building Models

The segmented image is an integral part of the SM method. As such, its accuracy needs to be evaluated. In this study, the segmented images were converted into the image-converted skymasks and were compared to their hand-labelled and generated counterparts, which can be seen below in Table 3. The hand-labelled skymask acts as a ‘control’ to compare the segmented image to, allowing evaluation of the CNN’s accuracy, while the generated skymask is generated at the ground truth using the 3D building model. As aforementioned, this ground truth is determined manually using Google Earth at the specific locations tested in this research. The former comparison evaluates the accuracy of the CNN, while the latter measures the discrepancy between the image-converted skymask and the skymask generated from the 3D model ground truth. 

The accuracy of the segmented skymask is compared in two ways, via mean difference (MD) and standard deviation difference (SDD), whose formulas are listed below. Ideally, the MD and SDD should both be zero when the image-converted skymask is compared to its hand-labelled and generated counterparts.
(10)$$\begin{array}{c} el_{*}\in \left\{el^{\mathrm{hand}\;\mathrm{labelled}\;\mathrm{image}},el_{GT}^{3D\;\mathrm{model}}\right\};\\ \mathrm{MD}=\bar{el^{image}}-\bar{el_{*}};\\ \mathrm{SDD}=\sqrt{\frac{{\left(el^{\mathrm{image}}-\bar{el^{\mathrm{image}}}\right)}^{2}}{360}}-\sqrt{\frac{{\left(el_{*}-\bar{el_{*}}\right)}^{2}}{360}} \end{array}$$

The MD between the segmented and hand-labelled skymask ranges from −6.32 to 1.42 degrees, while the SDD ranges from −0.69 to 2.38 degrees. Inaccuracies were largely the result of two sources; (1) overexposure of the image due to sunlight, and (2) failure to recognize reflective surfaces of glass buildings or buildings of similar color to the sky. These inaccuracies were especially prominent in position 3 shown in Table 2. This poses a challenge for the segmentation process, as these error sources are quite commonly encountered in the dense urban areas where the skymask matching is designed to be most helpful.

The MD between the segmented and generated skymask extends between −10.52 and 8.28 degrees, while the SDD was between −9.30 and 0.3. As mentioned earlier, the segmentation inaccuracy from the neural network contributed to the difference in MD and SDD. Other contributing factors include the assumption that the images are taken at mean sea level and inaccuracies in the 3D building model. During the experiments, the images might not exactly be at sea level due to handling difficulties. The change in position will decrease the size of buildings in the image and, therefore, reduce the elevation angle. Inaccuracies and low level of detail (LOD) in the 3D building model also affected the MD and SDD scores, as seen in Position 4 in Table 2.

These error sources can be mitigated in several ways. The segmented image accuracy could be increased by improving the convolutional neural network, which could help refine pixel classification and prevent the mislabeling of pixels. Overexposure could also be reduced by narrowing the aperture setting. To improve the accuracy of the generated image, an up-to-date 3D map with a higher LOD could be used. 

### 3.3. Positioning Results

Dots on the heatmap, also called a score map, represent the similarity between the candidates skymasks to the image-converted skymask, ranging from dark blue (0% similarity) to dark red (100% similarity). The heat maps are displayed below in Figure 4, with each diamond representing the positioning results of a different method. The positioning error of each method was then measured andrecorded in Table 4.

In Experiment 1, the results show the accuracy of the proposed skymask matching lags other methods in open-sky areas. A cumulative error increase of approximately 11 m from the smartphone initial position suggests that SM should not be used in open areas. Other processing methods increased the positional error by 1–2 m as well. This result was expected, as the lack of nearby structures meant that there were few building features to match. This is represented by the high similarity on most candidate skymasks in the open areas, where most candidate skymasks are deep red in color. There is a risk of increasing the along/across street error if the image-converted skymask is matched with a wrong skymask, demonstrated at this position. A simple workaround to address this problem is to disable skymask matching when the sky takes up more than 50% of the area in a sky-pointing image. In such situations, relying on the coordinates provided by the smartphone would yield better results as satellite measurements are likely in LOS to the receiver.

Experiment 2 is located in an urban-distinctive environment, which contains multiple distinctive high-rise buildings, which provided the features for image matching. When within these feature-rich environments, the skymask matching method improved upon the smartphone’s accuracy. The cumulative error is approximately 7 m for the SM method. Overall, SM improved positioning accuracy to an acceptable degree, and performed second best out of the post-processing methods, coming behind only 3DMA GNSS processing, which had a position error of approximately two meters. The inability of Conv and WLS to establish an accurate position was likely due to the nature of the highly urbanized environment. This environment, however, proved advantageous to SM, which had a bounty of distinctive features with which to match. 

Experiment 3 was categorized as an urban-complex type image. This meant that while buildings occupied a larger portion of these locations, resulting in poorer GNSS reception, SM also had to match complex features. Overall, the SM method yielded good improvements compared to smartphone GNSS solutions and WLS method. The cumulative error is about 11 m for the SM method, which is a 17 m improvement from the result obtained by the smartphone. While a noticeable improvement, the positioning accuracy still leaves something to be desired. It should be noted that SM had similar accuracy to 3DMA GNSS. Overall, these results suggest that SM can and should be considered being implemented in a complex feature environment and used in conjunction with other methods.

The 4th experiment belongs to the multiplex category, an environment with a balanced mix of trees, buildings, and other urban clutter. The fourth experiment also showed improvements in the accuracy of the positioning. Skymask matching reduced the positioning error from 48 to 15 m, with both along and across street error being substantially reduced. In this case, Conv performed the best with a cumulative error of under five meters, while SM came second with a cumulative error of about 15 m. SM’s positional accuracy was 1–2 m more accurate than WLS and 3DMA. One difficulty for existing positioning algorithms is the odd geometry of trees interfering with the GNSS signals. A future potential of skymask matching is to identify additional obstacles, such as trees and other urban clutter, that can be used for matching for positioning in a multiplex environment.

### 3.4. Heading Resolution Results

Before any images were taken, the camera (and thus the fisheye image) was aligned to north as described in Section 3.1. Skymask matching heading error and smartphone heading error for each image relative to the true north is shown in Table 5 (positive is clockwise).

The predicted bearing offsets for images 1, 2, 3, and 4 are all within acceptable parameters. The results show that, in an urban environment with features, the boundary of buildings can be used to accurately estimate the heading offset. Figure 5, below, compares the boundaries of the image-converted skymask and predicted candidate skymask. The boundaries of image-converted skymask is rotated/adjusted to the predicted bearing offset estimated by the algorithm used in this research.

All experiments show elevation angle discrepancies between the rotated image-converted skymasks and predicted candidate skymasks. In experiment 1, features were lacking; nonetheless, the few features were enough for SM to successfully gauge the orientation of the image-converted skymask by comparing to the candidate skymask. Experiment 2 shows high accuracy, likely due to the distinctive building boundaries in the image. The accurate prediction of the bearing offset, in turn, increased the position accuracy. Experiment 3’s high accuracy suggests that skymask matching can perform heading prediction well in an urban environment with complex building features. Experiment 4 shows large discrepancies to the candidate skymask elevation, likely due to 3D building model inaccuracy. Despite this challenge, the predicted bearing offset was only 1°. Hence, SM matching can be considered an accurate approach to estimate the heading of the user in an urban environment.

### 3.5. Discussion

Table 6 displays the limitations and assumptions made during this study. These factors will now be explained in further detail.

There are several limitations stemming from the usage of sky-pointing fisheye images. The first drawback comes from the location at which the fisheye images are taken. In this study, images are assumed to be taken at mean sea level. Due to human error in this experiment, the fisheye image may be slightly slanted in one direction or otherwise not taken correctly, negatively affecting accuracy because the building heights visible on the image will change. This problem can be mitigated when used for vehicular purposes, where the mounted sky-pointing fisheye camera will stay level.

Precautions must also be taken with the image datasets used to train the convolutional neural network. Since the network labels each individual pixel, the images must be highly accurate. This is a significant problem due to the hand-labelled nature of these images, human errors may result in inconsistent labelling, especially around objects like trees. This can largely be mitigated by setting strict guidelines on how to segment images. For example, in this project, trees were assumed to be solid objects, and any patches of sky visible between leaves and branches were ignored.

The semantic segmentation process also had some limitations. First, it requires many unique images in the dataset to increase variation and validation accuracy, there is the risk of overfitting on the images in the dataset. Another flaw is the limited number of classes, the current iteration of the code can identify only two classes, the sky and building classes. This means that objects that are not buildings, such as trees, signposts, and vehicles, are also classified as buildings. This also limits the usefulness of semantic segmentation, as different building materials are not identified. The former problem can be solved by labelling more unique images to increase the accuracy of the network. The latter problem can be solved by implementing more classes for objects, including building materials.

Another limitation comes from the inaccurate precomputed skymasks. This can occur in several ways: the 3D building model used to generate the skymask could be inaccurate, or the 3D model was out of date, new buildings could have been constructed, or old ones were torn down, leading to discrepancies between the fisheye image and skymask. The limitation can be solved by ensuring that the utilized 3D maps are highly accurate and constantly updated.

A limitation in skymask matching comes from the search radius of 50 m. If the convolutional GNSS position error is larger than 50 m, then the ability of the image-matching code to return an accurate result is limited by the search radius, reducing the ability of the code to return an accurate result. This could be easily offset by increasing the size of the search radius but would also increase the computing time and power required.

## 4. Conclusions and Future Works

This paper proposes a new method by introducing a new source of fisheye image data. First, a fisheye camera is used to capture the sky view fisheye image. Then, a convolutional neural network works in conjunction with active contouring to segment the fisheye image. The segmented image is then converted into a skymask and matched with its precomputed counterparts. The similarity between the image skymask and precomputed skymask is then regarded as the score of the position candidate. Compared with the 3DMA GNSS, the proposed algorithm can provide similar correction in an environment with distinctive building features. In addition, the heading angle estimated by the proposed skymask matching algorithm is very accurate.

However, the proposed skymask matching method still has limitations mentioned in Section 3.5 that must be addressed. The algorithm used in this research is also limited to comparing the building boundary only.

Several potential future developments are currently in progress, including the training of the CNN to distinguish between classes like trees and materials, not just sky and buildings. The image segmentation section of this method could also be repurposed due to its versatility. The ability to add new classes to differentiate means that given a large and high-quality dataset, the CNN can be adapted to a variety of different uses. An example is training the CNN to differentiate between different material classes.

Multi-class with larger amounts of detail provide the possibility for higher position accuracy. Allowing a distinction between these semantic classes opens further avenues to increase segmentation accuracy, as areas with vehicles and tree labels can be weighted less in scoring. While building materials can be factored into similarity matching to allow improved results in skymask matching. The skymask matching could also be further improved by extending the functionality to work in different weather, time, and brightness conditions. 

In light of the results presented in this paper, we conclude the skymask matching method provides degree-level heading accuracy in an urban environment, however, the positional accuracy is still a work in progress and requires further development, as it does not improve upon existing advanced 3DMA GNSS positioning solutions. Theoretically, if the method returned highly accurate positioning results, it would be well suited for vehicular navigation. This is due to the ease of integration into, and the stability of a vehicular platform. Integration into vehicles is simple because many modern automobiles are already equipped with cameras. As such, installation of a sky-pointing fisheye camera should be straightforward due to the preexisting infrastructure. Vehicles also make relatively stable platforms, which allows sky-pointing fisheye images to be captured with more accuracy than if the camera is held by hand.

## Figures and Tables

**Figure 1 sensors-20-04728-f001:**
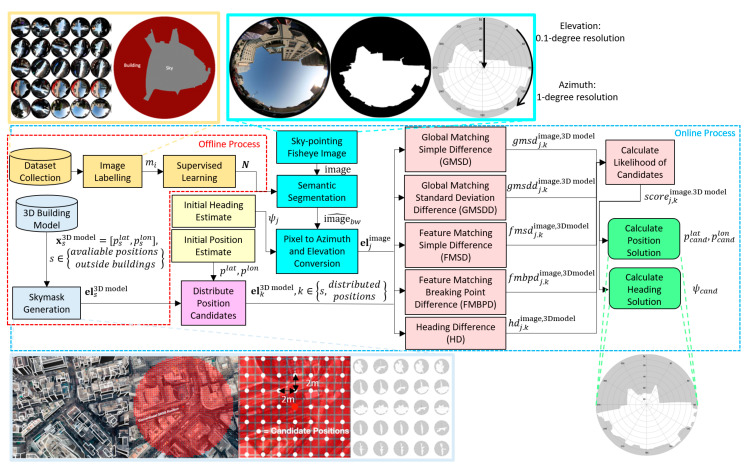
Flowchart of the proposed skymask matching based on images taken by sky-pointing fisheye camera and skymask generated by 3D building models.

**Figure 2 sensors-20-04728-f002:**
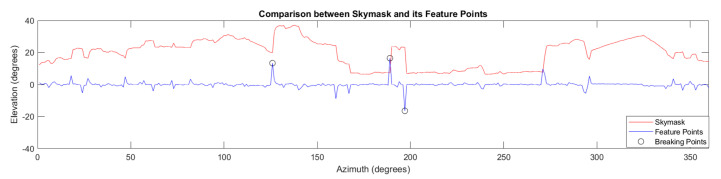
Comparison between the skymask and its feature points.

**Figure 3 sensors-20-04728-f003:**
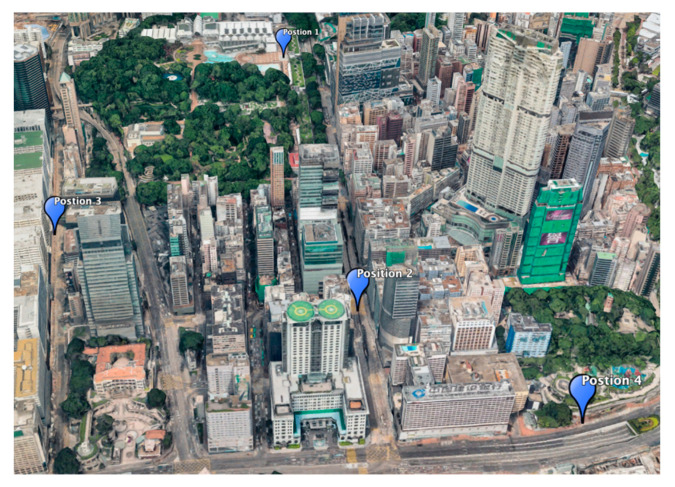
Experiment locations.

**Figure 4 sensors-20-04728-f004:**
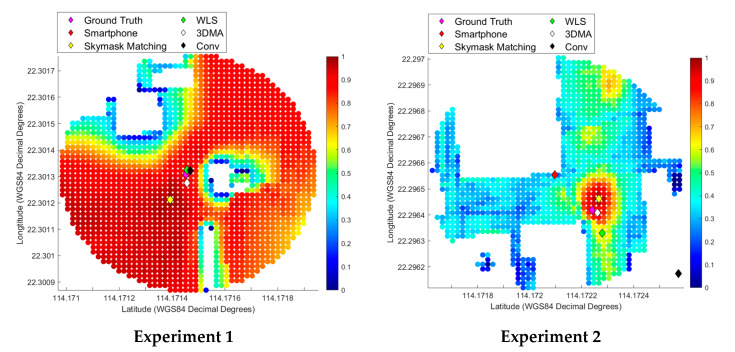
Heatmap on the similarity between the skymasks generated based on the fisheye images and 3D models based on the proposed skymask matching algorithm.

**Figure 5 sensors-20-04728-f005:**
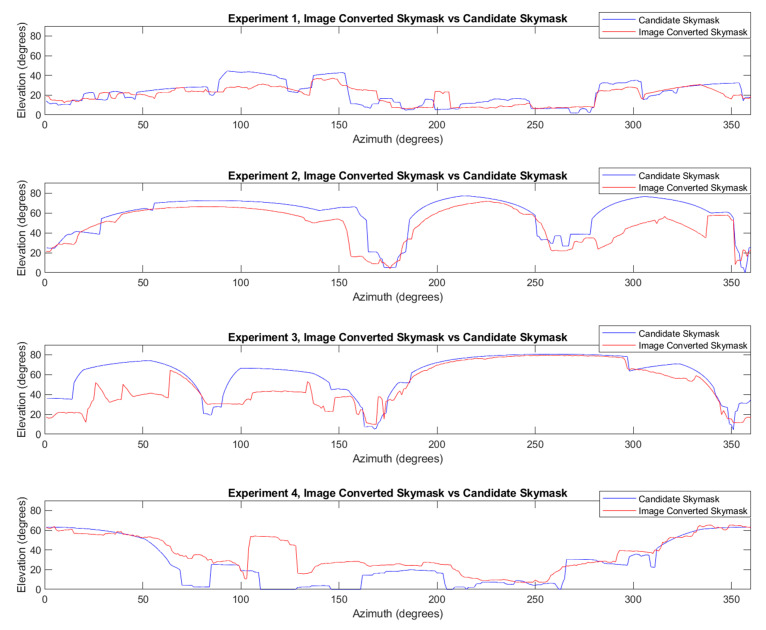
Azimuth and elevation comparison between the image-converted skymask and predicted candidate skymask.

**Table 1 sensors-20-04728-t001:** Fisheye image categories, and experiment locations.

Pos	Category	Environment	μ (Degree)	σ (Degree)
1	Clean	Few buildings and obstacles visible	18.18	8.44
2	Urban–Distinctive	High rise buildings, distinctive simple features	52.44	18.04
3	Urban–Complex	High rise buildings, complex features	56.14	18.84
4	Multiplex	High frequency of trees and other obstacles	36.65	18.29

**Table 2 sensors-20-04728-t002:** Close view of experiment locations 1–4 and the comparison between neural network segmented Images, labelled Images, their respective skymasks, and GT skymasks.

	Location	Fisheye Image	Neural Network Segmented Image	Hand Labelled Image	Neural Network Segmented Image-Converted Skymask	Hand Labelled Image-Converted Skymask	3D Model Generated GT Skymask
1	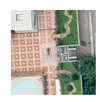	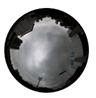	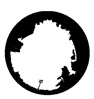	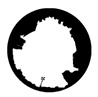	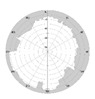	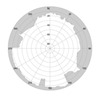	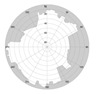
2	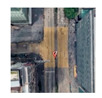	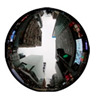	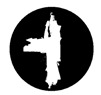	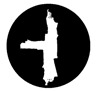	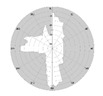	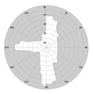	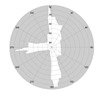
3	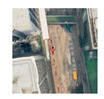	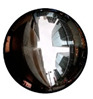	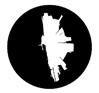	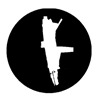	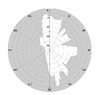	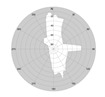	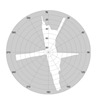
4	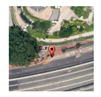	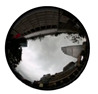	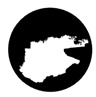	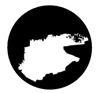	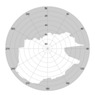	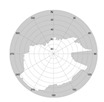	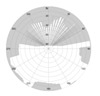

**Table 3 sensors-20-04728-t003:** Segmented skymask accuracy.

Comparison to:	Hand Labelled Skymask (degree)		Generated GT Skymask (degree)	
Position	Mean Difference	S.D Difference	Mean Difference	S.D Difference
**Position 1**	01.42	00.12	−04.02	−05.05
**Position 2**	−03.90	−00.22	−10.52	−01.93
**Position 3**	−06.32	02.38	−09.69	00.30
**Position 4**	−00.77	−00.69	08.28	−09.30

**Table 4 sensors-20-04728-t004:** Performance comparison of positioning results for different methods. Unit: meter.

Method		Smart Phone	WLS	Conv	3DMA	SM
Position	Error Type					
**Position 1**	Along-Street	00.77	01.96	01.26	03.83	10.39
	Across-Street	00.90	01.81	01.70	00.00	05.82
	Cumulative	01.18	02.67	02.12	03.83	11.91
**Position 2**	Along-Street	15.28	09.87	24.73	01.33	06.93
	Across-Street	12.13	02.52	31.66	01.82	00.85
	Cumulative	19.51	10.19	40.17	02.25	06.98
**Position 3**	Along-Street	04.30	10.59	09.33	09.97	09.30
	Across-Street	28.16	37.95	01.06	02.86	05.45
	Cumulative	28.49	39.40	9.39	10.37	10.78
**Position 4**	Along-Street	26.92	13.91	01.19	07.74	09.92
	Across-Street	41.33	06.79	04.46	14.87	10.33
	Cumulative	49.32	15.48	04.62	16.76	14.32

**Table 5 sensors-20-04728-t005:** Skymask matching heading error and smartphone heading error.

Experiment	Skymask Matching Heading Error (degree)	Smartphone Heading Error (degree)
**1**	−01	−08
**2**	00	08
**3**	00	−01
**4**	01	−45

**Table 6 sensors-20-04728-t006:** Summary of the limitations of the proposed skymask matching method.

Process	Assumptions/Limitations in this Experiment
Sky-pointing Fisheye Image	Assumes images taken from mean sea level.
	Assumes center of images are zenith pointing.
	Images were only taken during the day with sunlight.
Training Datasets	Requires a large dataset.
	Hand-labelled inaccuracy.
Semantic Segmentation	Risk of overfitting.
	Limited number of identifying classes.
Skymask Database	3D model might be outdated and/or imprecise.
Skymask Matching	Search radius.
	Compares only the building boundary.

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
