# Peer review of "Skymask Matching Aided Positioning Using Sky-Pointing Fisheye Camera and 3D City Models in Urban Canyons"

_sensors, 2020, doi:10.3390/s20174728_

Round 1

Reviewer 1 Report

The paper shows new method of a positioning for Global Navigation Satellite System (GNSS). The new method is supported by 3D-mapping-aided. Authors prove that proposed algorithm improves positioning performance in dense urban area. They analysed three different scenarios and they assessed obtained results by a comparison with other methods. I think about next scenario: lower and higher buildings. The lower buildings are near centre point of an image of fisheye and higher building are further. But a distance between the types of buildings is very long. In this case only higher buildings border on sky.

The paper is interesting but this is not good written. An analysing of results is very difficult - e.g. Table 3, 4 and Figure 3.  Below Table 4 the obtained results are analysed. In my opinion the results are not advantageous because the new algorithm does not give clearly better results from in comparison with other methods. The method has to be improve in the future. Now the effectiveness of this method is very poor.

Additionally during the analyse authors don't use the names of methods from page 8 (e.g. Allystar, BroadCom). The reviewer has to guess which method the Authors mean.

References aren’t written according to requirements of Sensors journal.

Author Response

We’d like to thank you for your time and feedback you’ve provided to us during the peer review. We have revised the paper to address the issues and other concerns that were raised. Please see the attachment for the point-to-point rebuttal letter. 

Reviewer 2 Report

This paper proposes a skymask matching algorithm based on sky-pointing fisheye camera and 3D city models. Then the author introduces the comparison method and carries on the experiment. The results were compared with other enhanced methods of positioning, and think that the method can provide similar correction in an environment with distinctive building features. This article innovation point is clear, argument is sufficient. I would like to point out the following observations:

  1. The link symbol is used in “3D-Mapping-Aided” in the first line of the abstract, but not uniformly in line 38.The author should examine the manuscript carefully to avoid similar situations of disagreement
  2. Line 187 "Figure" should not be italicized.
  3. The legend in Fig. 2 and Fig. 4 overrides the data in the figure.The author should change the location and size of the legend appropriately so that the reader can see the complete data.
  4. The format of references in this paper is disordered and inconsistent.For example, many articles are dated twice. The author should correct the references by referring to the style guide.
  5. The authors should point out the potential applications of their proposed method. For instances, what are the advantages in recording the images of the optical field modulation[J. Opt. A 6, 259 (2004); Applied Physics Letters, 2020, 116 (20): 201107; Nanophotonics 7, 677(2018)]?

In my opinion, the paper could be published if the authors were better able to respond these concerns.

Author Response

(The authors gave the same response as above.)

Round 2

Reviewer 1 Report

Thank authors for response. Now the paper sounds better and it includes more information about tests and analyses. The technical level of this paper is good. I hope that in this form the paper will found more readers.